# Centralization or Equalization? Policy Trend Guidance for Improving Grain Production Security in China

**DOI:** 10.3390/foods14060966

**Published:** 2025-03-12

**Authors:** Rongqian Lu, Xinhuan Zhang, Degang Yang, Yang Chen, Mingjie Cui

**Affiliations:** 1State Key Laboratory of Ecological Safety and Sustainable Development in Arid Lands, Xinjiang Institute of Ecology and Geography, Chinese Academy of Sciences, Urumqi 830011, China; lurongqian21@mails.ucas.ac.cn (R.L.); dgyang@ms.xjb.ac.cn (D.Y.); chenyang222@mails.ucas.ac.cn (Y.C.); cuimingjie19@mails.ucas.ac.cn (M.C.); 2University of the Chinese Academy of Sciences, Beijing 100049, China

**Keywords:** grain production, policy response, grain production and marketing areas, food security, China

## Abstract

Global grain production faces severe risks and challenges, such as the complex and volatile international situation and the frequent occurrence of extreme weather events. The effectiveness of national policies intended to support grain production security is becoming increasingly important. China has implemented a range of policies to promote grain production and increase the incomes of grain farmers. This study constructed a policy quantification model using a content analysis method to quantitatively analyze the various grain production support policies issued by the Chinese government. The application of sensitivity models and the difference-in-differences model to study the spatial response of China’s grain production to these policies enabled an assessment of the future policy trends of China, with the aim of enhancing grain production security. Grain production in the main grain-producing areas (MGPAs) responded best to the policy, the grain production–marketing-balanced areas (GPMBAs) were the second most responsive, and the main grain-marketing areas (MGMAs) responded to the policy to a lesser extent. The direct grain subsidy policy significantly contributed to an increase in grain production, especially in the MGPAs. The results of the study suggest that it would be more prudent for China’s future grain policy to guide the centralization of grain production toward the MGPAs. It will also be necessary to ensure that the government’s various support policies and subsidy funds are inclined toward the MGPAs, and the compensation mechanism should be improved to serve the interests of the MGPAs in terms of resources, ecology, and economy. This will help to enhance the comprehensive production capacity and production efficiency of the MGPAs, thereby guaranteeing China’s food security.

## 1. Introduction

The Global Food Crisis Report 2024 states that due to economic shocks, conflicts, and insecurity, as well as climate change and extreme weather events, nearly 735 million people were affected by hunger in 2023, including nearly 282 million people in crisis or more severe situations of acute food insecurity (IPC/CH Phase 3 or above). This represents an increase of 24 million people from the previous year, demonstrating the intensification of the global food crisis and the severity of food insecurity [1]. The stability of grain production is the foundation for ensuring food security. In recent years, the frequent occurrence of major emergencies, the increasing scarcity of global water resources, the growing demand for grain due to a rising global population, and the rapid development of industrialization and urbanization have increased the land area needed for grain production [2]. This presents numerous risks and challenges to global grain production [3,4]. Several countries have implemented different grain production support policies to stabilize grain production. The core content of U.S. agricultural policy is agricultural subsidies [5], which ensure farmers’ income and agricultural productivity through production subsidies, export subsidies, and fallow land programs. This has resulted in the improved competitiveness of the international market [6]. Russia has implemented an import substitution policy for agricultural products, conducts import restriction measures for agricultural products, constantly improves the self-sufficiency rate of agricultural products, and continuously increases the export of agricultural products [7,8]. Brazil has adopted a family agriculture support program and a low-carbon agriculture program [9], with a focus on supporting small-scale farmers and providing incentives to producers engaged in sustainable agriculture to promote grain production [4]. China has also implemented a large number of supportive policies to maintain high grain production and has achieved a high rate of self-sufficiency. With less than 9% of the world’s arable land and 6% of global freshwater resources, it produces nearly 25% of the world’s grain, feeding nearly 20% of the global population [10]. China’s grain production stabilized at over 1.3 trillion pounds for nine consecutive years from 2014 to 2023. It is recognized that China’s per capita grain possession will exceed 493 kg, which is higher than the 400 kg per capita international food security standard line. As China’s grain production support policy continues to evolve, the effective use of grain policy at different periods of development, given its guiding role, is not only a matter of food security for China’s population but also an important factor affecting the world’s food security [11].

In the more than 70 years since the establishment of New China, the country’s policies for ensuring food security have been continuously improved, with the implementation of a range of policies [12] and measures including land reform, the household contract responsibility system, the reform of the grain circulation system, the abolition of agricultural tax, and direct subsidies for grain farmers. This has ensured that overall grain production has maintained an upward trend. While classical economic theory suggests that increased grain production may lead to price decreases and could potentially lower farmers’ income, China has developed a comprehensive policy framework to address this challenge. The evolution of China’s grain policy reflects a careful balance between increasing grain production and protecting farmers’ income through price support mechanisms and subsidy systems. As a result, a development trend has gradually formed that initially focused on increasing grain production, then shifted to focusing on increasing the income of grain farmers, and finally, since 2004, shifted to a policy that has placed equal emphasis on increasing both grain production and the incomes of grain farmers [13]. Among the many policies implemented, the following are critical to guaranteeing food security in China. (1) The grain production and marketing zoning policy. In 2004, China formally designated grain production and marketing areas, dividing the country’s thirty-one provinces (autonomous regions and municipalities) into thirteen main grain-producing areas (MGPAs), seven main grain-marketing areas (MGMAs), and eleven grain production-marketing balanced areas (GPMBAs). The grain production and marketing zoning policy effectively guarantees China’s food security [14], with the MGPAs accounting for more than 78% of the country’s grain production in 2023. (2) The grain direct subsidy policy and price support system. In 1978, China began to implement a grain subsidy policy, initially only subsidizing the distribution chain. To protect farmers’ income while promoting production, the government established a minimum purchase price policy for major grains and a state grain reserves system to regulate market supply and price fluctuations. In 2004, a nationwide policy of direct subsidies for grain farmers was introduced, marking the shift of subsidies from the original distribution chain to the production chain [15]. In 2016, the direct grain subsidy was merged with the crop seed subsidy and the comprehensive agricultural subsidy to form the agricultural support and protection subsidy, which was later renamed the arable land fertility protection subsidy [16]. These subsidy policies have played an important role in stimulating farmers’ enthusiasm for growing grain. (3) The responsibility system for grain governors. China began implementing the governor responsibility system in 1994. In 2014, in response to the problems of the relaxation of grain production, neglect of grain distribution, and over-reliance on the central government in some places, the governor responsibility system for food security was implemented [17]. This system has highlighted the responsibility of local authorities for food security and further ensures that other food security policies are vigorously implemented.

China has implemented differentiated and focused grain production support policies for grain production and marketing areas to enhance its comprehensive grain production capacity. The regional distribution of grain production in China is currently highly concentrated, with the country’s grain production mainly occurring in the MGPAs, which are gradually being strengthened [18]. Studies have shown that in recent years, the MGPAs contributed nearly 78% of the total grain production, establishing them as the core of China’s grain production, playing a crucial role in guaranteeing China’s food security [19]. Although grain production in the GPMBAs has increased, their contribution to national grain production is limited [20], with their grain production accounting for about 18% of the total national production. In contrast, grain production in the MGMAs is on a downward trend, with their production capacity remaining low [21], accounting for only 4% of the country’s total. The increasing concentration of grain production in the MGPAs has led to significant regional challenges. These areas now face substantial grain surpluses that exceed their local consumption needs, requiring extensive storage facilities and efficient transportation systems to move grain to deficit areas. Additionally, this intensive production has intensified pressure on local agricultural resources and environmental carrying capacity [22,23]. This spatial mismatch between production and consumption zones creates logistical challenges and increases infrastructure demands in the MGPAs. The phenomenon of concentrating grain production in dominant areas or in a few MGPAs essentially indicates a shrinkage of these areas [24]. Under the existing conditions regarding agricultural resources, such as the level of grain production, the sown area, and the demographic structure of the workforce, the capacity for grain production increases in some MGPAs has nearly reached its upper limit [25]. A continued increase in grain production is therefore difficult to achieve [26,27]. While national policies effectively support grain production and farmers’ income, some MGPAs face a structural development dilemma. These regions, as the backbone of China’s grain production system, focus heavily on agriculture and produce large quantities of grain. However, due to the low added value of agriculture and the short industrial chains associated with grain production, these regions experience relatively weaker economic growth compared to the MGMAs, which have more-diversified economies. Furthermore, the limited fiscal revenue in the MGPAs restricts local governments’ ability to invest in infrastructure and public services, further constraining their economic development. As a result, the MGPAs have fallen into a development dilemma: “the more grain produced, the weaker the economy and the poorer the finances” [28]. This has led to a continuously widening gap in economic and social development between production and marketing areas, with a further imbalance in the regional layout of grain production [29]. Some researchers have argued that while optimizing and improving the compensation mechanism for the MGPAs, there should be a comprehensive promotion of their construction to continue leveraging their grain production advantages [30], stabilize grain production, enhance grain production capacity and efficiency, and provide more commercial grain production [31]. However, due to the conflict between grain production and economic development in the MGPAs, together with the increasingly severe international food risks [32,33], the Chinese government requires that the MGPAs, MGMAs, and GPMBAs all maintain their production areas and levels. It is also necessary to continuously improve the comprehensive grain production capacity in the MGPAs, effectively stabilize and enhance the grain self-sufficiency rate in the MGMAs, and ensure that the GPMBAs are fundamentally self-sufficient in grain [34]. Some researchers have suggested redrawing the boundaries of grain production and marketing areas, clarifying the grain production tasks for each zone, further optimizing the layout of these areas, implementing regional policies, clarifying responsibilities, and establishing and perfecting a multi-level benefit compensation mechanism [35]. This would allow the three major functional grain production areas to share the responsibility for China’s food security.

In recent years, sensitivity analysis has gained increasing attention as an important tool that supports uncertainty quantification and decision assessment. Research indicates that sensitivity analysis provides new perspectives for evaluating the effectiveness of decision-making schemes in policy formulation [36]. By employing sensitivity analysis, researchers systematically identify the impact of explanatory variables on model outcomes, offering unique insights into the complex dynamics of urban densification [37]. Furthermore, the application of sensitivity models in the agricultural sector is gradually revealing its potential, particularly in exploring the effects of changes in arable land area on food production and the sensitivity evaluation of food yield in response to changes in arable land area [38]. These studies not only enrich the theoretical foundation of sensitivity analysis but also provide empirical support for policy formulation. In the realm of policy quantification research, the increasing application of quantitative methods is driving advancements in this field. For instance, the application of the IIASA IAM framework provides essential context for the quantification of Shared Socioeconomic Pathways (SSP2s) [39]. Simultaneously, the development of comprehensive indicators has made it possible to assess the sustainability of the European agricultural food system, highlighting the significance of quantitative methods in policy implementation [40]. Additionally, comprehensive quantification studies on global forest fragmentation aim to guide the formulation of forest protection and restoration policies, further demonstrating the practical applications of policy quantification research [41]. In a study examining the effects of policy implementation, Guo et al. [42] employed a non-radial directional distance function to quantify performance and utilized a multi-period difference-in-differences approach to empirically investigate the impact of smart city pilot policies on energy and environmental performance. At the same time, the application of IFPRI’s global computable general equilibrium model MIRAGRODEP enables the simulation of the impact of agricultural subsidies on global greenhouse gas emissions, focusing on the combined effects of subsidies and border restrictions [43]. Furthermore, pilot policies for low-carbon cities significantly influence corporate pollution emissions, with researchers further assessing the policy effects of new energy demonstration cities on carbon emission efficiency through an SDID model that incorporates spatial lag terms [44,45]. The analysis of the impact of natural resource endowments and green finance on green economic efficiency has also provided empirical evidence for policy formulation [46]. These studies offer rich empirical support for the quantitative analysis of policy implementation effects. Despite some progress in the quantitative analysis of policy implementation effects, research on policy responses following implementation remains insufficient. Existing literature primarily focuses on assessing the direct impacts of policies, while in-depth discussions on actual implementation are relatively scarce. Particularly in the study of policy responses, most analyses remain at the qualitative level, concentrating mainly on coping strategies and methods and lacking systematic quantitative research. Therefore, it is necessary to explore the responses following policy implementation in greater depth.

There are two perspectives regarding the spatial pattern of grain production that can be derived from current studies. One suggests that the trend toward the regional concentration of grain production will become increasingly evident in the future [47], progressing toward regional specialization and the geographical division of labor [48]. This is expected to be beneficial for leveraging the comparative advantages of grain production across different areas, ensuring the steady growth of grain production [49]. The opposing perspective holds that the over-concentration of grain production could exacerbate its vulnerability [50], making it less resistant to the impacts of major epidemics or natural disasters [51], thereby threatening the stability of China’s grain production. Considering the significant influence of policies on China’s grain production [52,53,54], this study analyzed the spatial response of China’s grain production to the relevant policies that have been implemented. The aim was to determine whether the long time series dynamics of policy’s influence on grain production has tended more toward centralization or regional equalization. Here, ‘centralization’ refers to the concentration of China’s grain production tasks in the MGPAs, which would pool resources and have the advantage of providing sufficient commodity grains for the nation. In contrast, ‘equalization’ refers to the distribution of grain production responsibilities across the three major functional grain production areas, with each region undertaking a portion of the production task. The aim is to reduce production disparities between regions. In this study, we first quantified the various grain production support policies introduced by the Chinese government and analyzed the spatial response differences in grain production within each functional area and its constituent provinces. We then selected subsidy policies that promote the income of grain farmers and analyzed the differences in the response of grain production in the three major functional areas before and after the implementation of these policies. Through a comprehensive analysis of the responses of grain production to the implemented policies, we assessed whether future Chinese grain policies should prioritize spatial concentration or a more balanced distribution of production.

## 2. Materials and Methods

This study employed a comprehensive approach that combines qualitative and quantitative methods [55,56] to empirically investigate the implementation effects of China’s grain production support policies [57,58]. Our aim was to explore the responses of grain production to national policies and the impacts of national policies on grain production. The research is grounded in established empirical research methodologies, drawing on Ketokivi and Choi’s [59] perspectives on case study methods, which advocate for the use of multiple research methods to enhance the validity of research findings, as well as Noyes et al.’s [60] theory of integrated methodological design, which emphasizes the importance of integrating qualitative and quantitative data to provide a more comprehensive and in-depth understanding of research questions. The application of integrated methods in empirical research can bridge the gap between qualitative and quantitative research paradigms, offering more-comprehensive explanations for complex social phenomena. This approach enabled us not only to quantify the impacts of policies but also to explore the potential mechanisms and contextual factors affecting their implementation and effectiveness.

Based on this, we utilized a policy quantification model [61] to systematically analyze the intensity of various grain production support policies issued by the Chinese government from 1985 to 2023 across different dimensions. Based on the assessment results, we employed a sensitivity model [62] to study the responses of grain production to changes in policy intensity. Additionally, this paper uses a difference-in-differences model [63,64] to empirically evaluate the effects of the direct grain subsidy policy implemented in 2004. By leveraging these systematic analytical frameworks, our research aimed to provide empirical evidence regarding the interactions between national policies and grain production, ensuring that our findings are robustly supported by theoretical foundations and empirical data. This comprehensive methodology not only enhances the rigor of our analysis but also offers new methodological perspectives for agricultural policy research.

### 2.1. Data Sources and Processing

#### 2.1.1. Policy Data

The policy data used in this study primarily consisted of the various grain production support policies issued by the Chinese government from 1985 to 2023. This included laws, regulations, opinions, and notices at the national level that encourage and support farmers in grain production. The policy data were sourced mainly from government websites, such as the official Chinese government website, the Ministry of Agriculture and Rural Development, the State Administration of Grain and Material Reserves, and from policy statistics websites, including the Peking University Legal Information Database (https://pkulaw.com/, (accessed on 26 January 2024)). By examining the title, type, issuing authority, main content, and key policy measures of each document, a total of 145 policy documents related to grain production support were identified. Due to the extensive number of policies, Table 1 presents only a partial selection of the grain production support policies collected. Because some policies were declared invalid after a certain period, the study included only those policies that have been continuously effective since their promulgation. This ensured that the cumulative policy strength calculated in this study reflects the combined impact of all effective policies up to and including each year. Policies that were declared invalid after a certain period were excluded from the analysis.

#### 2.1.2. Food Data

The food data used in this study primarily encompassed grain production; the total area sown with grain crops; the effectively irrigated area; discounted agricultural fertilizer application; the area affected by a natural disaster; the number of agriculture, forestry, and fishery practitioners; and the per capita disposable income of rural residents across the 31 provinces in mainland China. These data were obtained from the national website of the China Bureau of Statistics (https://data.stats.gov.cn/, (accessed on 15 August 2024) and the China Statistical Yearbook for recent years. The descriptive statistics of each variable are shown in Table 2.

### 2.2. Research Methods

#### 2.2.1. Policy Quantification Model

We selected the policy content analysis method to analyze and study the various grain production support policies issued by the Chinese government [65]. Initially, we established a policy quantification model based on five dimensions [66]: policy level, policy goal, policy measure, policy power, and policy implementation. These dimensions are conceptually distinct and operationally defined with separate scoring criteria. However, in practice, they may exhibit some correlations, as higher-level policies often include more concrete measures or stronger implementation mechanisms. This interdependence is consistent with established policy analysis methodologies and ensures a comprehensive evaluation of policy strength. Subsequently, we used the Delphi method [67] to score the collected grain policy texts across these five dimensions, according to the policy quantification criteria outlined in Table 3. We consulted 15 experts in the Delphi method to score the policy texts. These experts were selected from academia, government agencies, and agricultural organizations to ensure a balanced representation of expertise. This scoring process allowed us to analyze the changes in the strength of grain production support policies in China. We used Equation (1) to calculate the policy strength of individual policies:(1)$$S_{ij}=L_{ij}\left(G_{ij}+M_{ij}+P_{ij}+I_{ij}\right)$$
where $S_{ij}$ is the policy strength of the j-th policy in year i; and $L_{ij}\;$, $G_{ij}\;$, $M_{ij}\;$, $P_{ij}$, and $I_{ij}$ are the policy level score, the policy goal score, the policy measure score, the policy power score, and the policy implementation score of the j-th policy in year i, respectively.

The annual policy strength of policies enacted each year was calculated according to Equation (2):(2)$$S_{i}=\sum_{j=1}^{n}S_{ij}$$
where $S_{i}$ is the annual policy strength in year i and n is the number of policies enacted in year i:(3)$$C_{i}=\sum_{i=1}^{i}S_{i}$$

Cumulative policy strength is the sum of the policy intensities of all effective policies in play before and in that year. The cumulative policy strength $C_{i}$ for a given year can be calculated from (3).

The calculation of cumulative policy strength is based on the condition that all policies included in the dataset have remained continuously effective since their promulgation. This is aligned with the policy selection criteria outlined in Section 2.1.1, which ensured that only policies with sustained effectiveness were included in the analysis. Policies that became invalid after a certain period were excluded to ensure that the cumulative policy strength accurately represents the combined impact of all effective policies over time.

The level of the policy refers to its importance and impact and is primarily determined by the issuing authority and the content of the policy code. The policy goal refers to whether the content of the policy directly reflects its specific goals or expected results, particularly in terms of its support for and promotion of grain production. Policy measure refers to the specific methods and means adopted by the government or relevant agencies in the policy document to realize the policy objectives and their degree of detail. Policy power refers to the expression and tone of attitude conveyed in the policy text, which is used to reflect the degree of urgency of the policy implementation requirements. Policy implementation refers to whether the content of the policy mentions which departments are specifically responsible for the implementation of the policy to ensure that it can be effectively implemented and whether the expected results were achieved.

#### 2.2.2. Sensitivity Model

Sensitivity is the degree of response of a system to changes in external conditions [68]. To investigate the response of grain production to various grain production support policies introduced by the Chinese government, a sensitivity analysis model was adopted from Liu et al. [38]:(4)$$\beta_{t+1}=\left[\left(G_{t+1}-G_{t}\right)/G_{t}\right]/\left[\left(P_{t+1}-P_{t}\right)/P_{t}\right]$$
where $\beta_{t+1}$ is the sensitivity coefficient of grain production to changes in the strength of China’s grain production support policy during the period from t to t+1; $G_{t}$ and $G_{t+1}$ are the grain production levels in the base and end periods, respectively; and $P_{t}$ and $P_{t+1}$ are the policy strengths of the grain production support policy in the base and end periods, respectively. If β≤0, grain production and policy strength are inversely related, and grain production is not sensitive to changes in policy strength, i.e., grain production is not good for China’s policy response. If β>0, grain production and policy strength are isotropic, and a change in grain production is influenced by a change in the magnitude of the policy strength, i.e., grain production is better for China’s policy response. Larger values of β indicate greater sensitivity of grain production to changes in policy strength, i.e., smaller changes in policy strength cause larger fluctuations in grain production, indicating that grain production responds better to grain production support policies.

#### 2.2.3. The Difference-in-Differences (DID) Model

The effects of certain key policies within grain production support policy can also indicate the response of grain production to these policies. The DID model was adopted to assess the impact of China’s direct grain subsidy policy [69]. The implementation of the 2004 direct grain subsidy policy was taken as a quasi-natural experiment, and a DID model was constructed to empirically test its effect on national grain production [70,71,72,73]. The period of 1998–2004 marked a significant phase for China in promoting the reform of the grain circulation system. Subsequently, from 2004 to 2014, the government of China introduced a series of major policy measures to bolster grain production, emphasizing the enhancement of the national food security guarantee system and the strengthening of agricultural support and protection mechanisms. Consequently, we used grain production data from 1998 to 2014 as our research sample. The year 2004, when the direct grain subsidy policy was implemented, was designated as the policy shock year. The MGPAs, which the Chinese government has focused support on, were designated as the experimental group, while the other 18 provinces served as the control group. The following DID model was then constructed:(5)$$Y_{it}=\alpha +\beta {DID}_{it}+\delta {control}_{it}+\mu_{i}+\gamma_{t}+\varepsilon_{it}$$
where i is province; t is time; Y is grain production; DID is the interaction term of the direct grain subsidy policy, i.e., the cross-multiplication term of period and treat (period is a time dummy variable [equal to 0 before 2004 and 1 in 2004 and later], and treat is a dummy variable for grouping provinces [equal to 0 for provinces not affected by the policy and 1 for provinces affected by the policy]); control is the set of all control variables; μ and γ are province-fixed effects and time-fixed effects, respectively; and ε is the random error.

The dependent variable was grain production, and the core independent variable was the direct grain subsidy policy interaction term (DID), calculated with the expression DID=treat×period. This is the product of the virtual variable for a province that implements the direct grain subsidy policy (treat) and the virtual variable for the duration of the implementation of the direct grain subsidy policy (period). To control the influence of other factors on grain production, the area sown with grain crops; effectively irrigated area; discounted agricultural fertilizer application; area affected by a natural disaster; number of agriculture, forestry, and fishery practitioners; and per capita disposable income of rural residents were selected as control variables. Due to the values of the variables differing greatly, a regression analysis was performed after calculating the logarithm of the variables to reduce the absolute difference between the data and avoid the influence of individual extreme values.

To ensure the credibility and robustness of the empirical results, a heterogeneity test, parallel trend test, and placebo test were also conducted to verify the results [74]. The heterogeneity test mainly explored the differences in the influence of the independent variables on the dependent variables for different regions or groups. A time trend graph can be used to determine the trend in changes between experimental and control group samples before and after the implementation of a policy, which is a prerequisite for DID research [75]. The placebo test can reduce the potential that the regression results are due to chance and exclude the possible effects of other policies or random factors on the differences before and after policy implementation [76]. To test the parallel trend hypothesis, drawing on the research methodology of Yang et al. [77,78,79], the event analysis method was used to estimate the effect of the implementation of the direct grain subsidy policy on grain production and construct the following specific model:(6)$$Y_{it}=\alpha +\sum_{k\geq -3}^{4}\varphi_{k}{DID}_{it+k}+\delta {control}_{it}+\mu_{i}+\gamma_{t}+\varepsilon_{it}$$
where k denotes the k-th year before and after the implementation of the policy. Data from 2001 to 2008 were used for testing, i.e., three years before and four years after the implementation of the policy. The coefficient $\varphi_{k}$ is the difference in grain production between the experimental and control groups in the k-th year after the implementation of the direct grain subsidy policy. The other variables in the formula are consistent with the model (5). The detailed estimation results and analysis of this model will be presented in the Section 3.

## 3. Results

### 3.1. Quantification of the Overall Policy for Grain Production Support

The Chinese government has long been committed to promoting grain production and ensuring the income of grain farmers and has continuously adjusted and improved its grain production support policy. This study quantified the grain production policy by establishing a policy quantification model by considering five dimensions—policy level, policy objective, policy measures, policy strength, and policy implementation—to objectively reflect the changes in the overall policy strength of grain production. The quantitative results (Figure 1) show that the annual policy strength of China’s overall grain production support policy has fluctuated but generally strengthened over time, with three phased peaks in policy strength in 1986, 2004, and 2021, respectively.

In 1985, the initial exploration of market-oriented reforms of the grain-purchasing and marketing system frustrated farmers’ incentives to produce. The overly rapid restructuring of agricultural production, coupled with the impact of natural disasters, resulted in a significant reduction in the area sown with grain and a decline in production per unit area, leading to a serious decline in grain production that year. To protect farmers and encourage them to produce and sell grain, and to ensure a smooth transition of grain policy to the next stage, the Chinese government formally introduced a “dual-track” reform policy of grain contract purchase and market bargaining in 1986. Grain production recovered after the implementation of the policy. From 1998 to 2003, the encroachment of arable land by urban development, the rapid development of cash crops, and the reduction in the grain-planting labor force led to a reduction in grain production year after year. To address these issues, the Chinese government proposed a policy through which industry would feed agriculture in 2004. This initiative involved a phased reduction in the agricultural tax rate, providing direct subsidies to grain farmers, subsidizing the purchase of high-quality seeds and agricultural machinery, and setting a minimum purchase price. This not only incentivized farmers to grow grain but also effectively increased the comprehensive grain production capacity, with production increasing by 38.77 million tons over the previous year. This represented the largest increase in grain production in China’s history. Since 2019, global circumstances have rapidly evolved, with the COVID-19 pandemic significantly affecting both the supply and demand dynamics of the grain market. To withstand the various risks and challenges that grain production may face, the Chinese government proposed a 35-item policy to strengthen agriculture and benefit farmers in 2021, including one-time subsidies for actual grain farmers. The central financial administration has explicitly supported a package of policies aimed at bolstering grain production, including storing grain on the land, storing grain in technology, and stabilizing subsidies for the planting of grain. The implementation of these policies has been effective in promoting the stable development of grain production and increasing the comprehensive production capacity of grain.

### 3.2. Spatial Response to the Overall Policy to Increasing Grain Production

Based on the quantitative outcomes of grain production policies implemented in China, a sensitivity model was applied to explore the effect of the spatial response of grain production in different functional grain production areas to the overall policy strength of increasing grain production. This enabled the regional implementation of these policies to be evaluated. Figure 2 compares the 1985–2004 and 2004–2023 time periods. The changes in the sensitivity of grain production in the MGPAs and GPMBAs in response to changes in the strength of the policy in the latter period was evaluated. It was found that the response of grain production in these two functional areas to the overall policy became increasingly effective over the study period. The response of the MGPAs was better than that of the GPMBAs, and the policy effectively promoted grain production in these two functional areas. However, the sensitivity of the MGMAs weakened over time, indicating that its grain production response to the overall policy was poor, and that the various grain production support policies played a limited role in grain production in the MGMAs. Based on the spatial response of grain production in different functional areas to the overall policy of increasing grain production during the two periods, not only were there regional differences in the response of the three major functional grain production areas to the overall policy of the Chinese government, but the response of the provinces within the different functional areas also varied.

#### 3.2.1. Response of the MGPAs to the Overall Policy of Increasing Grain Production

From 1985–2004 to 2004–2023, the sensitivities of provinces in the MGPAs displayed an increasing trend in both phases (Figure 3a), indicating that their grain production was responsive to the overall policy of increasing grain production. Given their superior conditions and foundations for grain production, the MGPAs bear the primary responsibility for maintaining the absolute security of the national grain ration and for safeguarding the priority rights to policy support and economic compensation. Consequently, the Chinese government’s grain production support policies are predominantly geared toward the MGPAs, resulting in the most notable policy impact occurring within these regions. In 2023, the MGPAs accounted for more than 78% of the country’s total grain production. However, while the overall response of the MGPAs has increased, there were also inter-provincial differences, with Heilongjiang and Inner Mongolia showing the best response, while Sichuan and Hunan had a relatively weak response. The grain-sown area in Inner Mongolia increased by 67.04% in 2023 compared to 2004, and grain production increased by 162.93% over the same period. In Heilongjiang, the grain-sown area increased by 74.31% and grain production increased by 159.52% in 2023 compared with 2004. The grain-sown area in Hunan increased by 0.20% in 2023 compared with 2004, while grain production increased by 16.21% over the same period. Conversely, the grain-sown area in Sichuan decreased by 1.12% in 2023 compared with 2004, and grain production increased by 14.21%. This reflects the relatively stable grain-sown area in Hunan and Sichuan, accompanied by improvements in grain yield due to factors such as technological advancements and optimized production practices. In contrast, provinces like Heilongjiang and Inner Mongolia exhibited significant increases in both sown area and production during the same period, driven by their abundant arable land, favorable climatic conditions, and strong government policy support. On the other hand, provinces like Hunan and Sichuan showed relatively lower increases in sown area and production due to geographic and structural constraints, including fragmented farmland, a focus on cash crops, and the impact of rural labor migration. The interprovincial differences in the response of the MGPAs to grain production policy reflect the different production potentials of the provinces in the MGPAs. The provinces with a better response significantly increased their grain production capacity in terms of area and production, whereas the provinces with a weaker response had limited potential to increase their production capacity.

#### 3.2.2. Response of the GPMBAs to the Overall Policy of Increasing Grain Production

From 1985–2004 to 2004–2023, the sensitivity increased in seven provinces in the GPMBAs and decreased in the other four provinces (Figure 3b). This indicated that grain production in the GPMBAs responded well to China’s overall policy of increasing grain production, although a few provinces were less responsive. The Chinese government requires the GPMBAs to stabilize the sown area and production of grain and to maintain the self-sufficiency rate in each province. Although the grain production conditions in the GPMBAs are not as good as those in the MGPAs, they are also actively implementing their national grain production objectives. In 2023, grain production in the GPMBAs accounted for 17.81% of the national total, and the grain self-sufficiency rate reached 100%. Although its contribution to national grain production was limited, the GPMBAs were able to maintain self-sufficiency. Among the 11 provinces in the GPMBAs, Xinjiang’s grain production had the largest response to various national policies, one that was even larger than that of some of the main grain producing provinces. In 2023, Xinjiang was ranked first in the country in terms of the increases in total grain production and sown area, with a grain self-sufficiency rate of 203.92%. The GPMBAs became the region with the largest amount of grain that can be transferred out, which was mainly influenced by China’s strategic requirements for Xinjiang, i.e., the Chinese government requires Xinjiang to take advantage of its rich reserves of arable land resources and large-scale farmland to create an important national supply base for high-quality agricultural and animal husbandry products. China’s grain production support policies were better implemented when they were overlaid with grain strategies, further reflecting the effect of strong policy guidance on China’s grain production.

#### 3.2.3. Response of the MGMAs to the Overall Policy of Increasing Grain Production

From 1985–2004 to 2004–2023, the sensitivity of two provinces in the MGMAs’ increased, while in the other five provinces, it decreased (Figure 3c). This indicated that grain production in the MGMAs responded poorly to China’s overall policy of increasing grain production, and that only a few provinces responded with a positive outcome to the policy directives. The MGMAs are relatively well developed economically, with generally high urbanization levels. Consequently, the per capita arable land area is much lower than the national average, and the policy goal in these areas has been to develop the economy as the main priority. This has led to non-agriculturalization and the use of arable land for activities other than growing grain, and therefore, the grain production capacity of the MGMAs is relatively low. In 2023, the grain production of the MGMAs accounted for only 4.3% of the country’s total grain production, and the grain self-sufficiency rate was 24.41%. The region had a relatively high degree of dependence on the MGPAs for its grain supply. The poor condition of arable land and limited incentives to grow grain have jointly contributed to the insensitivity of the MGMAs to the various policies introduced by the state to promote grain production, and the degree of response of their grain production to the policies was also low. Among the MGMAs, Tianjin had the best grain production response to the policy, with the grain production area and production doubling over 2004–2023. The sowing area increased in 19 provinces across the country during 2004–2023, but among the MGMAs, the only increases occurred in Tianjin. The grain sowing area increased by 47.99% and grain production increased by 108.3%, while the grain self-sufficiency rate reached 46.87% and the rations self-sufficiency rate reached 66.2%. If more importance is given to grain production in the MGMAs, there is some potential to increase the comprehensive grain production capacity. However, given that Tianjin’s grain sown area in 2023 was only 390,000 ha (26th in the country), the growth potential will remain limited.

### 3.3. Spatial Response of the Key Policy to Increase the Incomes of Grain Farmers

The DID model was applied to study the effectiveness of the direct grain subsidy policy that was implemented in 2004 and empirically analyze its impact on grain production. At the same time, the model was used to determine whether there were differences in the implementation of the direct grain subsidy policy in different functional grain production areas.

#### 3.3.1. Baseline Regression Results

An ordinary least squares regression of the effect of direct grain subsidy policy implementation was conducted according to model (5). The regression results are shown in Table 3. Column (1) shows the regression results without any control variables or fixed effects; column (2) shows the regression results with the addition of six control variables; columns (3) and (4) control for time-fixed effects on the basis of the previous two columns; column (5) further controls for province-fixed effects; and column (6) controls for both time-fixed effects and province-fixed effects, using robust standardized errors for both. The regression results showed that the implementation of the direct grain subsidy policy had a significant positive effect on grain production, and the model was better fitted after adding fixed effects. The significance level of the regression coefficient for the core independent variable in the difference-in-differences (DID) model improved from 10% to 1%, indicating that the direct grain subsidy policy has a significant positive effect. Column (2) of Table 4 shows that the estimated policy effect is 0.051, which implies that this policy has a notable impact. If the DID index increases by one unit, it will effectively promote a 5.1% increase in grain production.

For the control variables, under the influence of the direct grain subsidy policy, the coefficients associated with the area sown with grain crops; the effectively irrigated area; discounted agricultural fertilizer application; the number of agriculture, forestry, and fishery practitioners; and the per capita disposable income of rural residents were significantly positive, indicating that they played a positive role in promoting an increase in grain production. An increase in the grain-sown area directly increases grain production. Effective irrigation can ensure that the water needed for grain production is provided, the rational application of chemical fertilizer can improve soil fertility, and an increase in the number of people employed in agriculture, forestry, and fishery will ensure more labor inputs, which are factors that can significantly improve the efficiency of grain production. An increase in the per capita disposable income of rural residents usually implies an increase in farmers’ motivation to produce, an increase in investment in agricultural production, and a continuous improvement in agricultural infrastructure, all of which play a key role in increasing grain production. However, the coefficient of the area affected by natural disasters was significantly negative, indicating that it had a negative impact on grain production, and that natural disasters can directly affect the efficiency and scale of grain production, which in turn leads to a reduction in grain production.

#### 3.3.2. The Heterogeneity Test

To further determine whether there were differences in the role of different functional grain production areas due to the direct grain subsidy policy, a heterogeneity test was conducted from the perspective of the three major functional grain production areas. A group regression was conducted to study the impact of the implementation of the direct grain subsidy policy on the grain production of different functional areas. Table 5 shows the results of regional tests of the effectiveness of the direct grain subsidy policy. Column (1) shows the results without any control variables, and column (2) is the regression after adding the control variables. The results indicate that the implementation of the direct grain subsidy policy had a promotional effect on grain production in the MGPAs, with the results being significant at the 1% and 10% levels. There was no effect on grain production in the GPMBAs where the direct subsidy policy played a non-significant role, while the regression results of the impact on grain production in the MGMAs were significantly negative at the 1% level. This indicates that the implementation of the direct grain subsidy policy led to a significant increase in grain production in the MGPAs, while the impact on grain production in the GPMBAs was not significant, and grain production in the MGMAs was actually reduced. These results suggest that there were regional differences in the effectiveness of the direct grain subsidy policy, especially in the MGPAs, where the impact of the policy was most apparent.

#### 3.3.3. Robustness Test

The prerequisite for using the DID model is that the parallel trend test must be satisfied. As can be seen from the time trend graph Figure 4, before the implementation of the policy in 2004, the trend of changes in grain production in the MGPAs and non-MGPAs was roughly the same. However, after the implementation of the policy in 2004, a contrasting trend in grain production evolution between the two groups emerged. Grain production in the MGPA provinces increased steadily after 2004, whereas in the non-MGPA provinces, grain production displayed a downward trend. Therefore, it was preliminarily determined that the assumptions of the time-varying trends of the main and non-MGPA provinces before the year of policy implementation were basically satisfied. The differences in the trend lines after the year of policy implementation were then considered to be directly influenced by the grain subsidy policy.

Figure 5 presents the results of the parallel trend test conducted using model (6), which is designed to verify the common trend assumption for the implementation of the direct grain subsidy policy at the 95% confidence level. The figure shows that the coefficients $\varphi_{k}$ fluctuated around 0, and the 95% confidence intervals included coefficients with values of 0 before the implementation of the direct grain subsidy policy in 2004. This indicates that there was no significant difference in the grain production trends of MGPA and non-MGPA provinces prior to the policy implementation. After the implementation of the policy, the coefficients $\varphi_{k}$ became significantly positive and displayed a clear upward trend over time, indicating that the direct grain subsidy policy had a growing positive impact on grain production in MGPA provinces. This finding suggests that the implementation of the direct grain subsidy policy played a positive role in incentivizing the improvement of grain production in MGPA provinces. The gradual accumulation of policy effects over time further verifies the robustness of the benchmark regression results.

To avoid bias in the estimation results caused by unobserved variables, a placebo test was performed on the regression model. The experimental and control groups were randomly assigned to different groups, thus obtaining a randomly generated dummy variable for the experimental group. This was then replaced with the experimental group variable in the regression model and a further regression analysis was conducted to generate new estimated coefficients. This process was repeated 500 times to test whether there were any significant differences between the estimated coefficients and the baseline regression coefficients. The placebo test results are shown in Figure 6. The estimated coefficients obtained from random sampling were centrally distributed around 0 and followed a normal distribution. This was much smaller than the benchmark regression result of 0.248, indicating that there was no bias in the estimation results. The impact of the direct grain subsidy policy on grain production was therefore not affected by other unobserved factors. The placebo test was passed, which further verified the robustness of the benchmark regression result.

## 4. Discussion

### 4.1. Trends in China’s Grain Production Pattern Under Policy Guidance

Many studies have investigated the regional changes in grain production patterns in China [80,81,82], with the grain production capacity of the three major functional grain production areas showing a trend of polarization. The dominant position of the MGPAs in national grain production has been constantly consolidated [83], especially the concentration of MGPAs in the north. Grain production in the GPMBAs has increased, but the grain-producing areas have become less concentrated [84]. The MGMAs are experiencing a continuous decline in grain production, and their grain production capacity has diminished. The scale [85], efficiency, and comprehensive advantages of grain production [86] in the MGPAs are all greater than in other areas of the country, which has been achieved with the assistance of government policies [87]. Chai and Zhu [47] predicted that China’s grain production will display a trend of regional centralization in the future and gradually converge toward the regional specialization and regional division of labor. Some studies have also found that grain production faces threats from external risks, resource tightening, natural disasters, and price changes [88]. It is therefore necessary to not only maintain the stability of grain production in the MGPAs but also to promote grain production in the non-MGPA provinces. A balanced amount of production in each region is conducive to spreading the risk and effectively overcoming the various uncertainties in grain production [89]. Liu et al. [90] reported the need to develop an enthusiasm for grain production in different regions and to create conditions that enable regions to maximize their comparative advantages in grain production. This would ultimately realize the security of grain production from a global perspective.

This study examined the spatial response of the overall policy of increasing grain production and the key policy of increasing incomes for grain farmers. It was found that China’s grain production is currently exhibiting a trend of concentration in the MGPAs, and that the implementation of the various national policies to support grain production has been much better in the MGPAs than in the GPMBAs and MGMAs. Between 2004 and 2023, the share of national grain production in the MGPAs increased by 5.23%, while in the GPMBAs it decreased by 2.18%, and in the MGMAs it decreased by 3.05%. In terms of actual grain production growth rates during this period, the MGPAs demonstrated the highest growth rate, followed by the GPMBAs, while the MGMAs showed the lowest growth rate. This pattern reflects the varying regional advantages and constraints: the MGPAs benefit from favorable natural conditions and concentrated policy support; the GPMBAs maintain moderate growth through their balanced agricultural structure; while the MGMAs face challenges from urbanization pressures and less favorable cultivation conditions. This was a similar finding to that of Jiang and Wang [29], who reported that the pace of centralization of grain production in the MGPAs has accelerated, and the regional differentiation of grain production capacity has intensified. By analyzing the response of the different functional grain production areas to China’s grain production increase policy, it was evident that the overall response was most effective in the MGPAs. Some provinces in the GPMBAs exhibited a favorable response, whereas the majority of the MGMA provinces displayed a weaker response, with only a few regions showing a more positive reaction to the policy. This further confirmed the conclusion of Cao et al. [91], that in terms of their comparative contribution to grain production, the MGPAs made a significantly greater impact than the MGMAs and GPMBAs. In this study, we considered the 2004 direct grain subsidy policy as the key policy for promoting grain farmers’ incomes and investigated its spatial effectiveness. We found that the implementation of the direct subsidy policy could effectively promote an increase in grain production, especially in the MGPAs. This reflects the fact that there are significant regional differences in China’s grain production capacity and that the policy plays a pronounced role in guiding the MGPAs.

### 4.2. Trade-Offs Between the Centralization and Equalization of Grain Production in China

The centralization of grain production will integrate various resources and technological advantages in the MGPAs [92], and increase the input of infrastructure, capital, technology, and other production factors [93]. These factors are conducive to improving the efficiency of the agricultural environment, developing green and low-carbon agriculture [94], and realizing growth in grain production under the joint promotion of scale and efficiency [95]. However, the centralized production of grain will increase the intensity of development of soil and water resources and increase the use of fertilizers and pesticides. This will ultimately place pressure on the resources and environment of the MGPAs [96]. The centralization of grain production in the MGPAs has accordingly hindered the development of other high-profit industries. Additionally, the transfer of large quantities of grain out of the MGPAs has resulted in the loss of potential benefits to the MGPAs. This will affect the enthusiasm of farmers in the grain-growing provinces and regions and will also affect the long-term sustainability of the grain system [97]. The centralization of grain production in the MGPAs also concentrates the risk in the MGPAs [98], and in the event of a major natural disaster or major epidemic in the region, national grain production will be substantially reduced. The pressure on the supply of grain in the MGPAs will increase, which will eventually threaten national grain production [84,99].

The equalization of grain production can disperse the risks posed by the centralized production of grain in the MGPAs and ensure a stable grain supply. However, if the three major functional grain production areas are equalized to increase grain production capacity, i.e., if all areas are required to expand the area used for grain cultivation in accordance with the increased grain production target, provinces with unsuitable natural conditions for grain cultivation may face constraints in grain distribution, potentially resulting in decreased production efficiency [100,101]. Additionally, simultaneous expansion across all areas could lead to an oversupply of grain in the market, exerting downward pressure on output prices. This would not only reduce farmers’ returns but also increase their reliance on government subsidies, further exacerbating financial burdens and threatening the long-term sustainability of grain production. Moreover, by increasing grain production, each functional area will significantly increase China’s stock-to-consumption ratio, which currently exceeds the FAO standard level of 17–18%. Excessively high grain stockpiles can lead to a series of conflicts, such as difficulties in selling grain, pressure on the government to collect and store grain, and heavy financial burdens for farmers [102]. Because of the relatively high cost of grain production in China [103], grain farmers’ returns are low, and subsidies are needed to incentivize farmers to grow grain [104]. However, if the three major functional grain production areas expand their grain production areas at the same time, there will be a problem because the more grain production there is, the more government subsidies there will be, and the greater the financial difficulties will be [105].

The centralized and equalized approaches to grain production each present distinct advantages and disadvantages that require careful consideration. While centralization concentrates climate and epidemic risks in the MGPAs, equalization across all functional areas could lead to significant inefficiencies and financial burdens. After weighing these trade-offs, we suggest that a strategic division of responsibilities among the three functional areas would be more beneficial, where grain production is primarily concentrated in regions with natural advantages for cultivation. This approach advocates for “adapting measures to local conditions” rather than suggesting that land centralization is inherently superior to dispersion. The MGPAs, with their favorable conditions, would focus on grain production, while other functional areas would contribute according to their comparative advantages—whether in ecological preservation or diversified agricultural products. This division of responsibilities aims to optimize resource allocation, reduce overall production costs, and enhance efficiency while mitigating the risks through appropriate policy support and technological innovation. Such a strategy entails the efficient and rational allocation and use of resources to facilitate policy implementation, unify management, and ensure grain production security through a coordinated approach among all functional areas.

### 4.3. Suggestions for the Orientation of Future Grain Production Policies

(1) China should coordinate and optimize its pattern of grain production, adhere to the trend of centralization, and ensure the enhancement of grain production capacity in major production areas. At the national level, it is necessary to dynamically adjust the division of grain production and marketing areas, optimize the division of grain production and marketing areas at the provincial level according to their comparative advantages and potentials for grain production, and divide a certain number of grain-producing counties into county-level areas to implement grain production tasks at all levels. It is important to maintain the centralized production of grain in the MGPAs, push various national support policies and subsidies to the MGPAs with comparative advantages, improve the efficiency of grain production in the MGPAs, and accelerate the scale, intensification, and specialization of grain production in the MGPAs. The government should improve the incentives for grain cultivation in the MGPAs to fully leverage the benefits of grain production. It is crucial to prioritize the protection of the natural environment while producing grain and to ensure effective resource utilization, both of which will enable the sustainable development of grain production capacity and grain supply in the MGPAs in the future.

(2) The Chinese Government should increase its support for and protection of the MGPAs and improve the mechanisms for guaranteeing the incomes of grain farmers and for compensating the interests of the MGPAs. To stabilize grain production in the MGPAs, the government should continuously adjust and optimize its grain subsidy policy, increase the intensity of subsidies, improve the relevance and precision of subsidies for grain growing, safeguard the basic incomes of grain farmers, and enhance the incentivization of farmers in the MGPAs to grow grain. It should further improve the current vertical compensation mechanism, which is mainly based on financial transfers, and establish a horizontal compensation mechanism from local governments in the MGMAs to the MGPAs. The government should further advance the new model of multi-channel and diversified cooperation in grain production and marketing. Establishing a long-term cooperation framework involving risk and benefit sharing between the MGMAs and MGPAs is essential, with the MGMAs assuming the responsibility for compensating for the corresponding benefits. Moreover, it is crucial to fairly compensate for any profit losses caused by the development of grain production in the MGPAs.

(3) This study has proven that the implementation of various policies to increase production and income in China is effective and can promote the improvement of grain production, especially in the MGPAs, where the effect is more obvious. China’s future grain production will be concentrated in the MGPAs due to their substantial advantages in crop growth. It is imperative to prioritize the responsibility of the MGPAs for grain production, while ensuring that policy support is tilted to the MGPAs. Enhancing the benefits of the MGPAs through compensatory mechanisms is crucial for sustaining and bolstering grain production in these key regions. This trend of grain production in China has significance for other countries. China’s experience has shown that through the pooling of resources and policy support, centralized production is conducive to stable and increased grain production and can effectively improve the efficiency of grain production, as well as the level of regional specialization and scaling. This provides a feasible pathway toward solving the problem of global food security.

## 5. Conclusions

Based on the quantitative assessment of various grain production support policies issued by the Chinese government, this study explored the response of China’s grain production to both the overall policies that have been implemented to increase grain production and the key policies that have been implemented to increase the incomes of grain farmers. The best response to the overall policy of increasing grain production was observed in the MGPAs, followed by the GPMBAs, and then the MGMAs. The direct grain subsidy policy significantly promoted an increase in grain production, especially in the MGPAs, where the implementation of this policy was most effective. Although there were some specific problems with centralized grain production in the MGPAs, it did improve production efficiency and stabilize grain production. The balanced development of grain production in the functional grain production areas improved risk resistance, but efficiency and production levels could not be guaranteed, and there may even be more problems created, such as inventory backlogs. By analyzing the trade-offs between these two trends, we believe that the most appropriate approach is to guide China’s future grain policy toward the spatial concentration of grain production in the MGPAs. At the same time, there is a need to improve the compensation mechanism for the benefits of the MGPAs and the mechanism of guaranteeing the incomes of grain farmers. Increasing government policy and financial support for the MGPAs will bolster their comprehensive grain production capacity, ultimately safeguarding China’s food security. National guidelines on grain production policy development should take into account resource and environmental conditions, as well as the principles governing grain production. This will also provide a reference for other countries to weigh the centralized and balanced approaches to grain production guidance. 

However, this study also had some limitations. Because China’s policy framework is relatively large and the process of quantifying policies is complex, the study focused solely on national-level grain production support policies issued by the central government. The study did not consider local policies issued by each local government. Therefore, the study identified the response of each province to the implementation of Chinese government policies. In future studies, national and local policies should be considered together to identify the impacts of these different policies on grain production. Meanwhile, our future research should also consider employing two-stage least squares (2SLS) to more effectively address endogeneity issues, thus providing more reliable empirical analysis results. Additionally, future studies could explore more-complex production function models to delve deeper into the dynamic relationship between inputs and outputs in agricultural production, providing a more solid empirical foundation for policy formulation.

## Figures and Tables

**Figure 1 foods-14-00966-f001:**
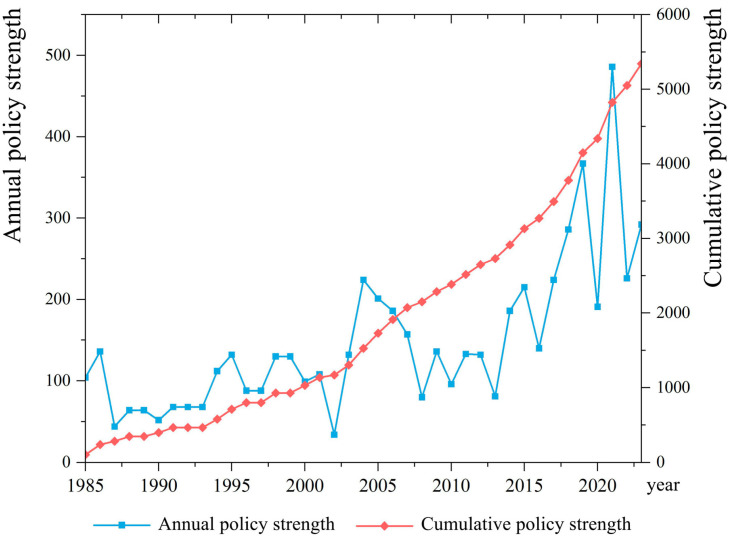
Changes in the strength of policy support for grain production in China.

**Figure 2 foods-14-00966-f002:**
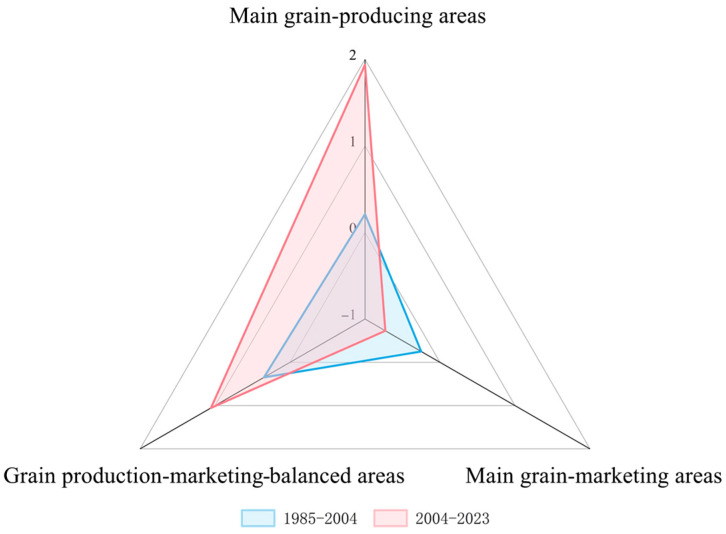
Spatial response of the three functional areas to the overall policy of increasing grain production.

**Figure 3 foods-14-00966-f003:**
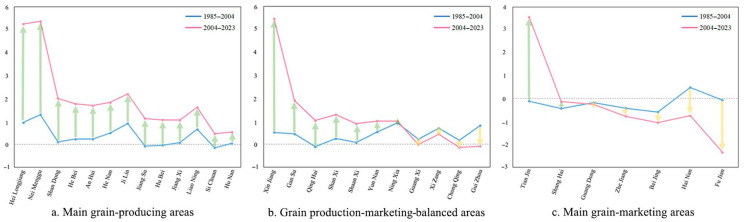
Response of provinces within the three functional areas to the overall policy of increasing grain production.

**Figure 4 foods-14-00966-f004:**
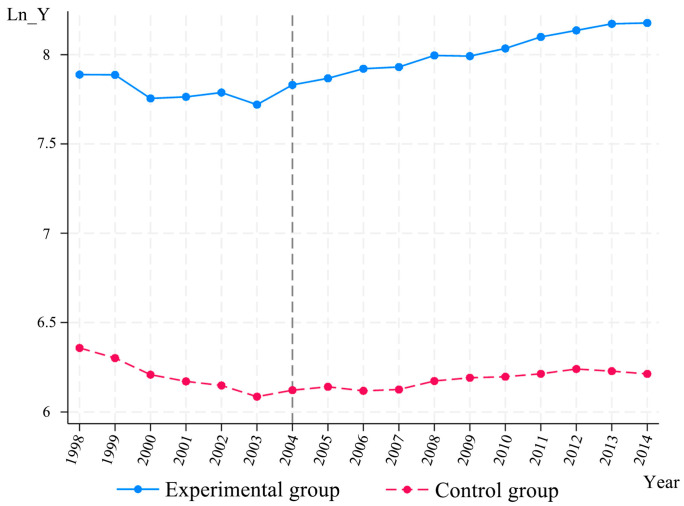
Time trend graph.

**Figure 5 foods-14-00966-f005:**
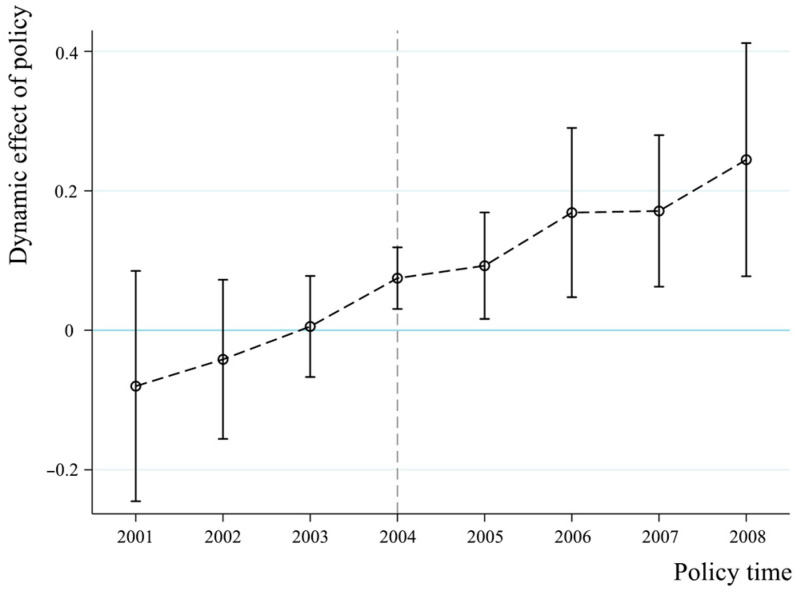
The results of the parallel trend test.

**Figure 6 foods-14-00966-f006:**
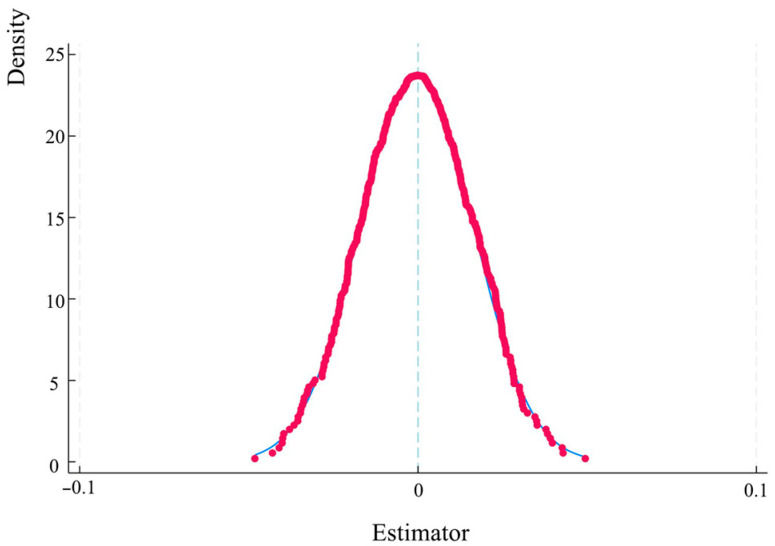
The placebo test results.

**Table 1 foods-14-00966-t001:** Grain production support policies issued at the Chinese government level (some examples).

Serial Number	Date of Issue	Title	Issuing Authority	Policy Content
32	2004	Opinions on a number of policies to further strengthen rural work and enhance comprehensive agricultural production capacity	Central Committee of the Communist Party of China, State Council	Focuses on stabilizing, improving, and strengthening policies to support agricultural development; further mobilizing farmers; and continuing to increase the implementation of policies such as the “two exemptions and three subsidies” strategy. Direct subsidies were introduced for grain farmers, and subsidies were introduced for farmers in some areas for the purchase of high-quality seeds and agricultural machinery. Direct subsidies will continue to be provided to grain farmers. The central financial authorities will continue to increase funding for subsidies for high-quality seeds and the purchase of agricultural machinery.
66	2014	Opinions on the establishment of a sound responsibility system for food security governors	State Council	Focuses on implementing and improving grain support policies. Grain subsidy policies should be carefully improved and implemented, and the precision and directionality of subsidies should be improved. New grain subsidies should be directed toward the main grain-producing areas and counties, and toward new grain production and management bodies. Supervision of subsidized funds will be strengthened to ensure that funds are provided in a timely manner and grain producers are fully subsidized. The agricultural insurance system should be improved and support for grain crop insurance should be provided.
145	2023	Law of the People’s Republic of China on Food Security	Central Committee of the Communist Party of China	Article 6: The Chinese Government establishes and improves the input mechanism for food security; adopts fiscal, financial, and other supportive policies to strengthen food security; improves the mechanism for the coordinated support of grain production, purchase, storage, transportation, processing and marketing; builds China’s food security industrial belt; and mobilizes grain producers and local people’s governments to protect their arable land, grow grain, and ensure food security. Additionally, it promotes the high-quality development of the grain industry and develops and enhances China’s ability to guarantee food security. Article 28: China will improve the mechanism for compensating the interests of the main grain-producing areas, perfecting the system of financial transfers to the main grain-producing areas and counties that produce large quantities of grain and mobilizing incentives for grain production.

**Table 2 foods-14-00966-t002:** Descriptive statistics regarding grain production in China.

Variables	Unit	Variable Symbol	N	Mean	SD	Min	Max
Region	Pcs	Region	527	16	8.953	1	31
Year	Year	Year	527	2006	4.904	1998	2014
Grain production	Ten thousand tons	Y	527	1686	1389	58.03	7404
Area sown with grain crops	Thousand hectares	Sown area	527	3514	2720	120.2	13,968
Effectively irrigated area	Thousand hectares	Irrigation	527	1849	1433	143.1	5342
Discounted agricultural fertilizer application	Ten thousand tons	Fertilizer	527	160.3	132.9	2.500	705.8
Area affected by a disaster	Thousand hectares	Damage	527	1353	1145	0	7394
Agriculture, forestry, and fishery practitioners	Ten thousand people	Manpower	527	955.0	747.3	24.75	3559
Per capita disposable income of rural residents	Yuan	Income	527	4997	3428	1231	21,192

**Table 3 foods-14-00966-t003:** Quantitative criteria for grain production support policies.

Dimension	Score	Standard for Judging
Policy Level	5	Issued by the National People’s Congress and its Standing Committee
	4	Issued by the Central Committee of the Communist Party of China and the State Council
	3	Issued by the General Office of the State Council
	2	Issued by ministries and commissions of the State Council
	1	Issued by the National Bureau, administered by ministries and commissions under The State Council
Policy Goal	5	Clearly defined policy objectives with clear methodological criteria
	4	Policy objectives are clear and unambiguous, but standards are vague
	3	Policy objectives are clearer
	2	Policy objectives are vague and not specifically articulated
	1	The policy vision is expressed only at the macro level
Policy Measure	5	List of specific measures, each with implementation and control criteria and their specification
	4	List of specific measures with more-detailed implementation and control criteria
	3	List of more-specific measures, categorized in a number of ways to give broad implementation elements
	2	List of some basic measures and a brief description of their implementation
	1	Only macro-level content, no concrete operational programs
Policy Power	5	Strong tone of voice descriptions such as “must”, “resolutely”, “forbidden”, “must not”, etc.
	3	Stronger descriptors such as “fully implemented”, “fully utilized”, etc.
	1	General tone descriptions such as “ensure”, “improve”, “strengthen”, etc.
Policy Implementation	5	There are multiple well-defined sectors for joint implementation
	3	There is a sectoral implementation
	1	No clear sectoral implementation

**Table 4 foods-14-00966-t004:** Baseline regression results.

Variables	(1) Ln_Y	(2) Ln_Y	(3) Ln_Y	(4) Ln_Y	(5) Ln_Y	(6) Ln_Y
DID	0.248 *	0.051 *	0.248 *	0.047 *	0.248 ***	0.248 ***
	(1.85)	(1.82)	(1.84)	(1.72)	(7.85)	(8.75)
Ln_Sown		0.759 ***		0.776 ***		
		(36.28)		(33.37)		
Ln_Irrigation		0.135 ***		0.138 ***		
		(5.98)		(5.77)		
Ln_Fertilizer		0.114 ***		0.091 ***		
		(5.89)		(4.48)		
Ln_Damage		−0.071 ***		−0.068 ***		
		(−6.48)		(−6.28)		
Ln_Manpower		0.054 ***		0.063 ***		
		(3.29)		(3.69)		
Ln_Income		0.104 ***		0.166 ***		
		(6.59)		(6.95)		
Constant	6.212 *** (62.39)	−1.189 *** (−7.15)	6.334 *** (39.83)	−1.727 *** (−7.58)	4.708 *** (50.16)	4.830 *** (49.11)
Year	NO	NO	YES	YES	NO	YES
Region	NO	NO	NO	NO	YES	YES
Observations	527	523	527	523	527	527
R-squared	0.523	0.987	0.527	0.988	0.982	0.986

Notes: Robust t-statistics in parentheses. *** *p* < 0.01, * *p* < 0.1. Year denotes time-fixed effects and Region denotes province-fixed effects.

**Table 5 foods-14-00966-t005:** Heterogeneity test results.

VARIABLE	MGPA Ln_Y		GPMBA Ln_Y		MGMA Ln_Y	
	(1)	(2)	(1)	(2)	(1)	(2)
DID	0.248 *** (3.58)	0.051 * (1.82)	0.037 (0.44)	0.046 (2.47)	−0.318 *** (−3.41)	−0.053 * (−1.65)
Control	NO	YES	NO	YES	NO	YES
Year	YES	YES	YES	YES	YES	YES
Region	YES	YES	YES	YES	YES	YES
Constant	7.000 *** (178.87)	−1.189 *** (−7.15)	7.000 *** (160.64)	−1.845 *** (−11.37)	7.000 *** (211.03)	−1.959 *** (−11.85)
Observations	527	523	527	523	527	523
R-squared	0.333	0.987	0.221	0.984	0.354	0.985
Number of cities	31	31	31	31	31	31

Notes: Robust t-statistics in parentheses. *** *p* < 0.01, * *p* < 0.1. Control denotes the set of control variables described earlier, Year denotes time-fixed effects, and Region denotes region-fixed effects.

## Data Availability

The original contributions presented in the study are included in the article/Appendix A. Further inquiries can be directed to the corresponding author.

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
