# Peer review of "Centralization or Equalization? Policy Trend Guidance for Improving Grain Production Security in China"

_foods, 2025, doi:10.3390/foods14060966_

Round 1

Reviewer 1 Report

Comments and Suggestions for Authors

Centralization or equalization? Policy trend guidance for improving grain production security in China

Authors: Rongqian Lu et al.

The authors constructed a policy quantification model through a content analysis method to quantitatively analyze the various grain production support policies issued by the Chinese government. They apply sensitivity models and the difference-in-differences model to study the spatial response of China's grain production to these policies enabled an assessment of the future policy trends of China, with the aim of enhancing grain production security.

The reviewer did not find obvious novelty in this paper.

Abstract. The important data results (Grain production in the main grain-producing areas (MGPA) responded best to the policy, the grain production−marketing balanced areas (GPMBA) were the second most responsive and the main grain-marketing areas (MGMA) responded to the policy to a lesser extent) are expected.

Grain production data in China is well documented, as well as the theoretical framework and  methodology of data analysis. The discussion of the data is comprehensive and the results are likely to contribute to policy trend guidance for improving grain production security in China. However, there is no novelty in the topic studied, there are several recent papers published on the subject, perhaps more relevant would be the study of different regions in China.

Reviewer 2 Report

Comments and Suggestions for Authors

See attached file.

Comments on the Quality of English Language

The quality of the English language is high and understandable. However, I think that a proofreading is necessary for the revised version.

Reviewer 3 Report

Comments and Suggestions for Authors

Article presents aim of research, that is: "determine whether the long time series dynamics of policy influence on grain production has tended more toward centralization or regional equalization". This research aim is clear and coherent with performed research. Is difficult to affirm that research is innovative or not, because theoretical review is insufficient. Theoretical review is very short and is presented in introduction text, what is not appropriate. Theoretical review must be more robust and long, and should be presented after introduction text, before materials and methods. In the beginning of materials and methods is lacking information about classification of the research, based on authors of methodology. 

I suggest to create text presenting state-of-art of scientific literature about this issue. Must detail effort to find articles in bases such as Scopus, Web of Science, and so on, presenting results of this research. Materials and Methods is well described and treatment of data was done correctly. Interpretation os results must be based on theoretical review.

My suggestions to improve article are: 1) create specific topic of theoretical review; 2) write first paragraph or two in materials and methods to present type of research was performed, based on, at least, two authors of methodology.

Round 2

Reviewer 1 Report

Comments and Suggestions for Authors

The authors made the suggested changes, answering in detail and completely 

Author Response

Comments: The authors made the suggested changes, answering in detail and completely.

Response: Thank you for your thorough review of our paper. We greatly value the valuable feedback you provided in the first round, and we have carefully revised each suggestion. Your constructive comments regarding the paper's innovations and the novelty of the research theme have significant academic guidance for us in deeply contemplating the research's value.

We sincerely appreciate the professional insights and valuable time you have dedicated to improving our paper. Your keen academic insight and rigorous scholarly attitude not only help us to examine our research work more comprehensively but also inspire us to continually enhance the academic quality and theoretical contributions of the paper. Through this round of in-depth revisions and improvements, we believe that the academic value and research depth of this manuscript have been substantially elevated.

Once again, thank you for your support and guidance throughout the review process. We are very grateful for the time and effort you have invested in providing detailed feedback, which has made a significant contribution to the refinement of our manuscript.

Reviewer 2 Report

Comments and Suggestions for Authors

The submitted paper is very well revised. There are few commnets.

The results show that sign of a variable is wrong. This is difficult to explain. I hope the author will revise the estimation results.
